# Evaluation of the Multiplex Real-Time PCR DermaGenius^®^ Assay for the Detection of Dermatophytes in Hair Samples from Senegal

**DOI:** 10.3390/jof8010011

**Published:** 2021-12-24

**Authors:** Mouhamadou Ndiaye, Rosalie Sacheli, Khadim Diongue, Caroline Adjetey, Rajae Darfouf, Mame Cheikh Seck, Aida Sadikh Badiane, Mamadou Alpha Diallo, Therese Dieng, Marie-Pierre Hayette, Daouda Ndiaye

**Affiliations:** 1Laboratory of Parasitology and Mycology, Cheikh Anta Diop University, Avenue Cheikh Anta Diop, Fann, Dakar BO 16477, Senegal; khadim4.diongue@ucad.edu.sn (K.D.); mamecheikh.seck@ucad.edu.sn (M.C.S.); aida.badiane@ucad.edu.sn (A.S.B.); therese.dieng@ucad.edu.sn (T.D.); daouda.ndiaye@ucad.edu.sn (D.N.); 2Laboratory of Parasitology, Aristide Le Dantec University Hospital, Dakar BO 5005, Senegal; mamadoualpha.diallo@ucad.edu.sn; 3Department of Clinical Microbiology, Center for Interdisciplinary Research on Medicines (CIRM), University Hospital of Liege, 1-B-4000 Liege, Belgium; r.sacheli@chuliege.be (R.S.); aadjetey@chuliege.be (C.A.); rajae.darfouf@chuliege.be (R.D.); mphayette@chuliege.be (M.-P.H.); 4National Reference Center for Mycosis, University Hospital of Liege, 1-B-4000 Liege, Belgium

**Keywords:** dermatophytosis, hair, real-time PCR, DermaGenius^®^, Senegal

## Abstract

For the successful treatment of dermatophytoses, especially tinea capitis, there is a need for accurate and rapid diagnostic methods. A lot of recent literature has focused on the detection of dermatophytes directly on sample material such as nails, hair and skin scrapings. Molecular tools offer the ability to rapidly diagnose dermatophytosis within 48 h. This study aimed to compare the results of a commercial real-time PCR (real-time PCR) assay DermaGenius^®^(DG) 2.0 complete multiplex kit with those of conventional diagnostic methods (direct microscopy and culture). A total of 129 hair samples were collected in Dakar (Senegal) from patients suspected of dermatophytosis. DG was applied for the molecular detection of *Candida albicans*, *Trichophyton rubrum/soudanense*, *T. interdigitale*, *T. tonsurans*, *T. mentagrophytes*, *T. violaceum*, *Microsporum canis*, *M. audouinii*, *Epidermophyton floccosum*, *T. benhamiae* and *T. verrucosum*. Dermatophytes species and *C. albicans* were differentiated by melting curve analysis. The sensitivity and specificity of the PCR assay were 89.3% and 75.3%, respectively. DG PCR was significantly more sensitive than culture (*p* < 0.001). DG PCR is fast and robust to contamination. In this paper, the main questions discussed were the replacement of culture by a broad-spectrum fungal real-time PCR and the implementation of DG PCR into a routine laboratory in Senegal.

## 1. Introduction

Dermatophytes are keratinophilic fungi that are responsible for the majority of superficial fungal infections (dermatophytosis) and that lead to a reduced quality of life and a heavy economic burden for those affected [1].

Based on the most recent introduced taxonomy, this fungal group consists of more than 50 species distributed in the genera of Trichophyton, Microsporum, Epidermophyton, Nannizzia, Arthroderma, Lophophyton, Paraphyton, and Guarromyces [2].

Routine procedures for dermatophyte species identification rely on macroscopic examination of the culture such as pigmentation of the surface and reverse sides, topography, texture, and rate of growth, as well as on microscopic morphology: size and shape of macroconidia and microconidia, spirals, nodular organs, and pectinate branches. Further identification characteristics include nutritional requirements (vitamins and amino acids) and temperature tolerance, as well as urease production, alkaline production of bromocresol purple medium, and in vitro hair perforation [3]. Morphological and physiological characteristics can frequently vary. In fact, phenotypic features can be easily influenced by outside factors such as temperature variation, medium and chemotherapy, and therefore strain identification is often difficult. Furthermore, this system of identification is time-consuming and may be challenging for non-experts in morphology differentiation. Additionally, even the same strains may show diverse colony morphologies, making the identification of the causative organism more difficult [4].

Several novel molecular techniques have recently been developed for the rapid and accurate identification of dermatophytes [5]. A commercial multiplex real-time PCR assay for direct detection of fungi, particularly dermatophytes, in clinical material—in skin, nail scrapings and hair—has been evaluated in regard to their specificity and sensitivity for the identification of dermatophytes [6]. The DermaGenius^®^(DG) 2.0. complete multiplex kit (PathoNostics, Amsterdam, The Netherlands) is a new commercial real-time polymerase chain reaction (PCR) assay. The evaluation was based on the melting curve analysis. A total of 11 dermatophytes and one yeast fungus (Candida albicans) can be identified with the real-time PCR test at the species level. DG is a very fast and easy to perform test, especially since no post-PCR step is necessary.

This study aimed to compare the DermaGenius^®^ 2.0 PCR assay with KOH microscopy in combination with culture for diagnosis of clinically suspected tinea capitis.

## 2. Materials and Methods

### 2.1. Study Design

This is a cross-sectional descriptive three-year study (2016 to 2019) that was carried out on hair samples collected from patients clinically diagnosed with tinea capitis (TC) after dermatological consultation. The sampling, microscopy and culture procedures took place in the Laboratory of Parasitology and Mycology in Aristide Le Dantec University Hospital (LPM/ADUH) in Dakar, Senegal.

For each patient (one sample per patient), hairs were sampled from alopecic plaque areas by scraping with a sterile scalpel blade. This was preceded by the use of Wood’s 70 lamp which orients to certain species such as *M. canis* and *M. audouinii*. Then, loose hairs and scalp scrapings were divided in two parts. One part was used for the conventional diagnosis. The other part was kept sterile at ambient temperature for downstream PCR analyses. All 129 samples were divided into two groups: 73 positive KOH microscopy and negative culture, 56 positive KOH microscopy and positive culture. Then, the 129 samples were submitted to the complete multiplex real-time PCR DermaGenius^®^ (Pathonostics, NL) for the molecular detection of pathogenic dermatophytes plus *C. albicans* at the National Reference Center for Mycosis, University Hospital of Liège, Belgium.

### 2.2. Conventional Diagnosis

Diagnosis of TC was established at the LPM/ADUH in Dakar, on the basis of mycological examination including direct microscopy and culture as previously described [7]. Microscopic direct examination of all specimens was carried out in 20% KOH mount. All specimens were cultured on two plates/tubes, one containing Sabouraud dextrose agar (SDA) supplemented with chloramphenicol (Bio-Rad, Paris, France), and the other one containing SDA supplemented with chloramphenicol plus cycloheximide (Bio-Rad, France). Cultures were incubated at 25–30 °C and evaluated for growth after 48 h and then once weekly for a month. Positive specimens for dermatophytes were identified according to three criteria: growth rate, macroscopic and microscopic characteristics of colonies and sometimes on biochemical characteristics such as a urease test [8,9].

### 2.3. DNA-Extraction from Hair Samples

The QIAamp DNA Mini Kit 250 (Qiagen, Hilden, Germany) was used for DNA isolation from hair samples. First, a pre-treatment of the sample was done. A limited amount (<10 hairs) of short hairs (2 cm from scalp) with follicle/skin attached were completely submerged in a sterile 1.5 mL reaction tube containing 475 µL of ATL buffer and 25 µL of proteinase K and incubated overnight at 56 °C at 1000 rpm on a thermoshaker (Eppendorf, Paris, France). Then, after a brief centrifugation, 10 µL of internal control (PathoNostics, NL) was added and the mixture reheated for 1 min at 65 °C. 200 µL of ethanol were then added to the sample which was mixed on a vortex, centrifuged, and then loaded onto a provided spin column (Qiagen, Hilden, Germany). The spin column was centrifuged at 6000 rpm for 1 min and then placed on a new collection tube. The other tube containing the filtrate was discarded.

After two washes with buffer AW1 and AW2 (500 µL), the DNA was eluted in 100 µL volume with buffer AE. The extracted DNA was stored at 4 °C (if PCR is performed the same day, otherwise at −20 °C).

### 2.4. DermaGenius^®^ PCR

The PCR protocol for the DermaGenius^®^ (DG) 2.0 Complete multiplex kit (PathoNostics, the Netherlands) was performed on the LightCycler 480 II (Roche, Switzerland) according to the manufacturer’s instructions. Reagents for performing two separate multiplex PCR procedures were used: Master Mix 1 (MMX1) contained the originally designed specific PCR primer pairs and detection probes for *C. albicans*, *T. interdigitale*, *T. mentagrophytes*, *T. tonsurans*, *T. violaceum*, and *T. rubrum/soudanense*, and MMX2 contained the originally designed primer pairs and probes for *T. benhamiae*, *T. verrucosum*, *M. canis*, *M. audouinii*, and *E. floccosum.* The internal control (IC) is supplied as a M13 bacteriophage solution and is used to discriminate true negative samples from false negative samples, which can be a result of nucleic acid degradation, problems with the extraction protocol, PCR inhibition or test failure.

DNA samples were added to both MMX1 and MMX2 to a final volume of 25 μL and placed in the LC480 thermocycler. The enzymatic reaction was programmed as follows: 2 min at 95 °C followed by 45 cycles of 15 s at 96 °C and 60 s at 55 °C. The melting curve profile consisted of 2 min at 96 °C (hold) and melting at 45–85 °C. The duplex MMX1 and MMX2 reactions were run simultaneously in the same instrument but in separate wells. All PCR products were analysed by their melting temperatures. Positive controls and negative template controls (NTC) were included in each PCR run. Data analysis was performed using the 2nd-derivative and Tm-calling function of the LC480 software (version 1.5.1.62 SP2).

### 2.5. Statistical Analysis

Data were checked, entered and analyzed using Epi info version 7.1.5 (CDC, Atlanta, GA, USA) for data processing. The following statistical methods were used for analysis of results of the present study. Data were expressed as number and percentage for qualitative variables. A chi-square test (X^2^) was used to find associations between row and column variables. The agreement between different laboratory methods in measuring the same variable was estimated by Cohen’s kappa test (K). Sensitivity, specificity, positive predictive value (PPV), negative predictive value (NPV), and accuracy of different laboratory methods was determined. 

When comparing the techniques, total agreement statistics (in percent) were calculated as well as the kappa coefficient in the case of discrepancies between the two methods. The kappa agreement level was interpreted as follows: K < 0.20 poor, 0.21–0.40 Fair, 0.41–0.60 moderate, 0.61–0.80 good, and 0.81–1.00 very good [10]. For all statistical tests, the threshold of significance was fixed at 5% level (*p* < 0.05).

## 3. Results

Out of the 129 included patients, 30 (23.26%) were males and 99 (76.74%) females. Patient’s age varied from one to 80 years, along with a mean age of 23.64 ± 17.61 years. Figure 1 shows the distribution of TC according to age groups.

The patients in the age group between 0–10 years had the highest distribution (42.65%), followed by those between the ages of 11 and 20 years (25%). Patients whose ages were between 31 and 40 years (8.82%) had the lowest prevalence. Table 1 summarizes the strain information with direct microscopy, culture and DG PCR results.

Of the 129 patients clinically suspected of TC, 46.41% (56/129) were positive and 56.59% were negative (73/129) in culture. Dermatophytes were detected by DG PCR in 52.71% (68/129). This study shows that DG PCR has 89.3% sensitivity, 75.3% specificity, 81.4% accuracy, 73.3% PPV and 90.2% NPV (Table 2).

The DG PCR assay was more sensitive than culture for dermatophyte detection in patients (*p* < 0.05). The kappa coefficient in case of discrepancies between the two methods was good (k = 0.62).

The isolated dermatophytes were *T. soudanense* in 35 (27.1%) cases by culture and 51 (39.5%) cases by DG PCR, followed by *M. audouinii* in 18 (14%) and 17 (13.2%) cases and *M. canis* detected in 3 (2.3%) and 6 (4.7%) cases by culture and DG respectively. Two mixed infections *T. soudanense/M. audouinii,* 5 (3.9%) cases and *T. soudanense/M. canis,* one (0.8%) case were detected by DG PCR and not detected by culture (Table 3).

A total of 16 species detected by DG PCR as *T. soudanense* were negative in culture. Out of the five mixed infection *T. soudanense/M. audouinii*, culture isolated *M. audouinii* alone in four cases and *T. soudanense* alone in one case. Concerning the *T. soudanense/M. canis* mixed infection, *T. soudanense* was only isolated by culture. In this study, the others microorganisms in this test panel (i.e., *C. albicans*, *T. interdigitale*, *T. mentagrophytes*, *T. tonsurans*, *T. violaceum*, *T. benhamiae*, *T. verrucosum*, and *E. floccosum*) were not detected.

Figure 2a,b shows the melting curves of *T. soudanense* and *T. violaceum* species (positive control) as well as *M. audouinii* and *M. canis* species (positive control). The delineation between species was clear.

## 4. Discussion

TC is a major problem in dermatology because of its widespread distribution. In the present study, we compared the efficiency of a multiplex commercial real-time PCR with mycological cultures for the diagnosis of 129 clinically suspected cases of TC diagnosed in Dakar Hospital.

Concerning gender and age distribution, TC was confirmed in patients aged from one to 80 years with an average of 23.6 years. The highest prevalence of TC was found in the age group 0–10 years followed by 11–20 years. Females (76.47%) were more infected than men (23.53%). All these findings are in accordance with what was previously reported from the same laboratory [11,12]. Like in other studies performed in Nigeria [13], the prevalence of TC with respect to the age was lower for the age group 5–10 years (42.6%) than that of 11–15 years (50%). In this study, it was observed that the prevalence of TC in prepubertal females (56.7%) was higher than that of the males (35.3%) while the prevalence of TC among the pubertal age range of 11–15 years was higher in males (48.4%) compared to females (20%). In another study reported in Nigeria, Yemisi and al, conducted a case-control study to identify the causative agents and factors that predispose school pupils to TC [14]. They found that TC is more prevalent in children between the ages of four and seven years. These results support the suggestions that dermatophytosis, especially TC, is predominantly a pre-pubertal disease. Some of the explanations that have been accorded to this are that the fatty acids in the sebum produced at puberty may have some fungistatic properties, thereby preventing the infection of older children [15]. Another factor that may support the higher prevalence of TC among younger children is the likelihood of poor hygiene in pre-pubertal stages, compared to older children who usually become more conscious of their hygiene practices when they reach their teenage years [13,16]. Various conflicting views exist regarding the sexual predominance of TC which may be attributed to hair. Dressing and styling practices such as tight hair braiding, shaving of the scalp, plaiting, and the use of hair oils may promote disease transmission. However, the precise role of such practices remains a subject of study [17,18,19].

Hay et al., in 2017 [20] showed most affected patients are children of six months to 10–12 years of age. The clinical appearance of ringworm of the scalp is variable, depending on the type of hair invasion, the level of host resistance and the degree of inflammatory host response. TC can sometimes occur in adults and in this case is usually caused by anthropophilic dermatophyte species. The pattern varies from a few, broken-off hairs with little scaling, detectable only on careful inspection, to a severe, painful, inflammatory mass or kerion covering most of the scalp. Itching is variable. In all types, the characteristic features are partial hair loss with some degree of inflammation. In man, there is a correlation between inflammatory responses, T-lymphocyte activation, and recovery. In TC, the development of delayed-type hypersensitivity and presumed T cell mediated immunity to dermatophyte antigen correlates with recovery from the infection [21].

In this study the DG PCR test showed that two anthropophilic dermatophytes species, *T. soudanense* and *M. audouinii*, were the main etiological agents isolated from in TC followed by M. canis. These findings were similar from those observed elsewhere in Africa, more precisely in West Africa. *T. soudanense* is endemic throughout Africa [22]. *T. soudanense* was the most prevalent dermatophyte found in a previous study on tinea capitis in Dakar 2008 and 2013 [12], as well as in Mali [23]. Also, in this precited study, it was followed by *M. audouinii,* and *M. canis*, both preceded by *T. rubrum* [12].

In terms of polyparasitism, there were more coinfections detected by real-time PCR assays compared to culture method. In fact, two mixed infections by *T. soudanense/M. audouinii* and *T. soudanense/M. canis* were detected by DG PCR. Previously, a case of mixed infection has been reported in Senegal [24]. Concerning this case, at the first time, the culture of nail samples yielded *Microsporum langeronii (M. audouinii*). Due to the fact that this species is not a common agent of onychomycosis, the repetition of the sampling plus hair samples revealed tinea capitis due to *M. langeronii* in mixed infection *Trichophyton soudanense* after more than three weeks of incubation. Moreover, given its cultural characteristics, with a speed of increase slower than *M. langeronii (M. audouinii*), which has colonies extensively capable to mask any other dermatophyte, this justifies the fact that this species appeared secondarily during culture. Dermatophytoses due to two different dermatophytes are very rarely reported. These mixed dermatophyte infections, rare in humans, on the other hand appear to be well documented in pets, especially dogs and cats according to Mihaylov et al. [25].

Real-time PCR is a particularly attractive diagnostic method, because the amplification and detection of the pathogen DNA is performed in one step.

The DG PCR assay was shown to be highly sensitive and specific for the detection and identification of dermatophytes directly from clinical material. In our study the sensitivity and the specificity of the PCR were 89.3% and 75.3% respectively when microscopy and culture were considered as the gold standard. 

First, data on clinical samples tested with the DermaGenius^®^ 2.0 complete kit have been analysed, a poster presentation during the International Society for Human and Animal Mycology (ISHAM) 20th Congress in Amsterdam, The Netherlands [26]. The results of the kit have not been compared here with the results obtained from fungal culture, but with molecular analysis using PCR ELISA and ITS sequencing. For this study, samples were pre-selected, which included both frequent and less frequent and rare species of dermatophytes. Of 49 samples, the DermaGenius^®^ 2.0 Complete Assay 37 species was detected on specific melting temperature. In 12 samples, one dermatophyte species was detected, but closely related species were not differentiated.

Similar findings with 80% sensitivity and 74.4% specificity were reported in a retrospective investigation conducted with the DG PCR test (DermaGenius^®^) applied on nails. DG PCR performance was not different from histology combined with culture (*p* = 0.11) but the best diagnostic efficacy (88.4%, 122/138) was obtained by combining histology and DG PCR [27].

Other commercial kits which utilize this technology are on the market.

The Dermatophytes kit from Bio Evolution is a real-time PCR which is a further development of the PCR-ELISA Kit Onychodiag (formerly distributed by Bio Advance) [28]. The kit contains only one universal dermatophyte detection; however, the differentiation of genera or species is not possible. A study from the year 2016 proves the superiority of the molecular method in terms of the sensitivity in comparison to the culture. This study is the first retrospective evaluation of BioEvolution’s real-time PCR kit, which was carried out on 180 nails, divided into optimal and non-optimal samples. When comparing the number of dermatophytes found by culture and the molecular method, a larger number of dermatophytes was detected with this molecular kit only 23.3% (21/90) and 16.7% (15/90) respectively of the optimal and non-optimal samples, obtained from the same patients were found positive in culture, whereas the PCR resulted in 34.4% (31/90) of positive cases whatever the sample quality [29].

DG PCR and a recently developed microarray test (EuroArray Dermatomycosis, Euroimmun Lübeck, Germany) (EuroArray) were evaluated regarding their diagnostic specificity to identify dermatophyte DNA [6]. The EuroArray Dermatomycosis is a PCR-based procedure for detection of 56 fungi species causing infections of skin, hair and nails. Out of these 56 pathogens, 23 dermatophytes, three yeasts and three mould species can be identified.

In 2019, Uhrlab et al. compared DG PCR and EuroArray tests regarding their diagnostic specificity to identify dermatophyte DNA. The comparison of the two test systems shows that the microarray Dermatomycosis is much more specific, for which the test is carried out in two stages (PCR with subsequent hybridization). Evaluation at the scanner is quick and easy. The most common dermatophytes, but also rare species, are recognized. There are few incorrect identifications. Using the EuroArray, 22 out of 24 dermatophyte species were correctly identified. DG PCR can detect considerably less dermatophyte species than the EuroArray Dermatomycosis and does not include general dermatophyte detection. Pathogens frequently found in practice are covered in the best possible way by DG. For example, Euro Array does not permit the detection of *T. soudanense*. Our study shows that this agent is one of the main dermatophytes implicated in tinea capitis in Senegal but also around the world [30].

The requirement of this DG PCR test, which is used in daily practice (or routine) for the distinction between the anthropophilic and zoophilic species within the *T. mentagrophytes/T. interdigitale* complex issues should still be improved, also with a view to the importance of the different sources of infection. The real-time PCR has nine of 11 dermatophyte species correctly recognized. *T. soudanense* and *T. rubrum* will not be differently identified. This difference could be explained by the genetic match or close relationship of the African dermatophyte species with *T. rubrum* this species.

However, DG PCR is also faster in implementation, as no post-PCR step becomes necessary. The evaluation of the test results on the basis of the melting curves (Light-Cycler 480 II) is costly under demanded real-time PCR experience [6].

A considerable number of in-house real-time PCR techniques have been developed for the diagnosis of dermatophytes. They are not standardized, and in the vast majority of cases are used only by individual, or very few, laboratories involved in routine diagnostics. Many of these approaches are able to identify up to six taxa and dermatophytes in general [31]. A real-time PCR method directed against ITS1 for use with skin, nails and hair was compared with conventional methods by Wisselink et al. [31]. The real-time PCR showed a sensitivity of 97%, representing a significant increase in the detection rate for dermatophytes in clinical samples compared with the culture. Also, Bergman et al. reported the development of a single-tube dermatophyte-specific qPCR assay based on ITS1 sequences that allows the rapid detection and identification of 11 clinically relevant species within the three dermatophyte genera Trichophyton, Microsporum and Epidermophyton in nail, skin and hair samples within a few hours [32].

Recently, Walser and Bosshard developed a two-tube pan-dermatophyte PCR assay using six molecular beacon (SMB) probes. The first PCR uses dermatophyte-specific primers and enables detection and identification of most dermatophyte species. The second PCR with pan-fungal primers allows further differentiation of *T. interdigitale* and *T. mentagrophytes /T. quinckeanum*, *T. violaceum* and *T. soudanense*, and *T. tonsurans* and *T. equinum*, and detection of non-dermatophytes molds. The test was evaluated on 306 clinical specimens in comparison with microscopy and culture. Sensitivity and specificity of PCR for detection of dermatophytes in clinical samples were estimated to be 96.9% and 90.4% for culture 46.7% and 98.7%, and for microscopy 91.4% and 84.0%, respectively. The detection of non-dermatophytes molds by PCR and culture did not correlate [33]. Table 4 summarises various studies comparing commercial kits versus in-house real time PCR techniques. The sensitivity and specificity of our study is more or less equivalent to those of Hayette & al who worked on nails.

In this study, DG PCR allowed a clear separation between *T. soudanense* and *T. violaceum*. They can be genotypically similar from each other. In 2018, Nenoff et al., showed that for differentiating *T. soudanense* and *T. violaceum*, confirmation and refinements using other genes are needed. The family tree or dendrogram, based on the ITS1 and ITS2 region sequences, showed the phylogenetic differences of *T. soudanense, T. violaceum* and *T. rubrum*. A clear distinction is possible. Nenoff et al., also showed that sequencing of the translation elongation factor 1-α (TEF1-α) gene for a distinction between *T. soudanense* and *T. violaceum* was possible. On the other hand, the TEF1-α gene would not allow a distinction between *T. soudanense* and *T. rubrum*. This shows the specificity of DG PCR compared to ITS and TEF1-α sequencing to differentiate the complex *T. rubrum/soudanense/violaceum* [34]. It is important to notice that while DG PCR allow a clear distinction between *T. soudanense* and *T. violaceum*, no discrimination regarding the melting curves are possible regarding *T. rubrum* and *T. soudanense*. However, the clinical context of the infection is clearly different between these two species, *T. rubrum* being implicated in nails infections while *T. soudanense* is mainly responsible for TC. However, for skin samples, ITS PCR or conventional methods such as cultures and microscopy are mandatory to discriminate between both species. The limitations of our study would be to include some negative microscopy and negative culture specimens as “confirmed negative controls”, These will exclude any positive (i.e., false negative) cases that were not detected by both culture and DG PCR.

DG PCR can be used as rapid test when a clinician needs a fast and precise diagnostic.

It can also replace the ITS sequencing (which requires four to five days before obtaining results and needs a culture step) in order to obtain a faster dermatophyte identification in case of a doubtful microscopic examination of the culture. Finally, it is the convenient method to use when a patient is already under antifungal treatment, making the culture an inappropriate diagnostic method. Also, the DermaGenius^®^ multiplex PCR assay is the first real-time commercial PCR assay that combines the detection and differentiation of *T. rubrum* and *T. interdigitale* directly in nails, in addition to *C. albicans* [27]. Indeed, real-time PCR is an easy-to-use molecular method in laboratory because the detection of amplicons is automatically recorded by a dedicated software module associated to the thermocycler without the need of post-PCR steps. Furthermore, the results are easy and rapid to interpret for technicians trained in molecular biology. Up to now, only one other commercial real-time PCR assay has been validated for the detection of dermatophytes in nails that is also applicable for hair and skin specimens [29].

## 5. Conclusions

DG PCR showed excellent performance characteristics for the detection of dermatophytes and is significantly faster than culture techniques, which makes it very promising for routine diagnostics of dermatophytosis in Africa, and particularly in Senegal. It can help the clinician in initiating prompt and appropriate antifungal therapy. This technique is not only rapid but also simple and cheap in comparison to other molecular methods for the detection of dermatophytes.

## Figures and Tables

**Figure 1 jof-08-00011-f001:**
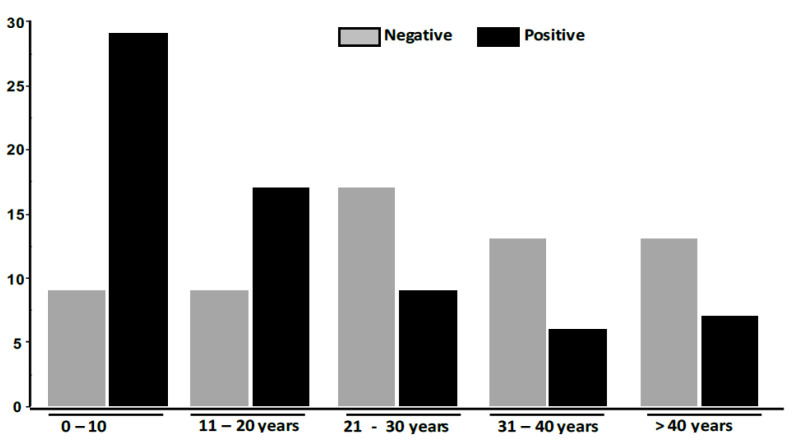
Distribution of *Tinea capitis* according to age groups.

**Figure 2 jof-08-00011-f002:**
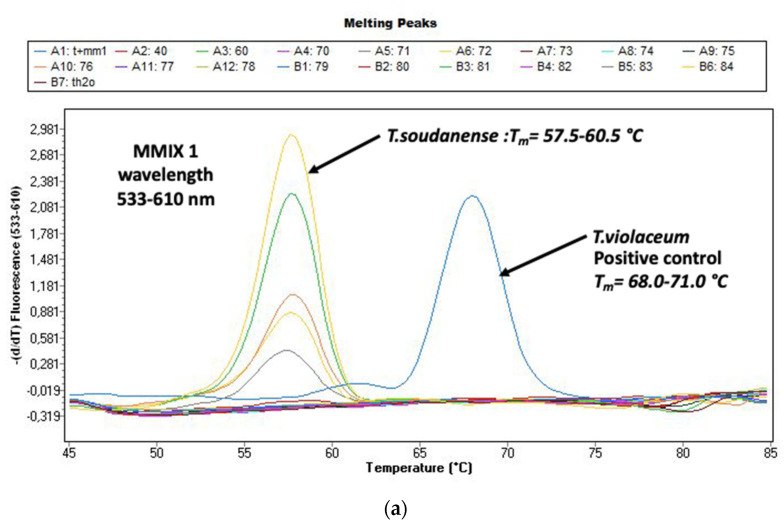
(**a**) Evaluation of the melting curves, seen on the LightCycler 480 II with the real-time PCR (polymerase chain reaction) (DermaGenius^®^2.0). Delimitation *Trichophyton* (*T.*) *violaceum* and *T. soudanense* is possible. (**b**) Evaluation of the melting curves, seen on the LightCycler 480 II with the real-time PCR (polymerase chain reaction) (DermaGenius^®^2.0). Delimitation *Microsporum (M.) canis* and *M. audouinii* is possible.

**Table 1 jof-08-00011-t001:** Strain information, with a summary of microscopy, culture, and PCR DermaGenius results.

Strain No.	Direct Microscopy	Culture	PCR DermaGenius
1	Endothrix	negative	Negative
2	Endothrix	*T. soudanense*	*T. soudanense*
3	Endothrix	*T. soudanense*	Negative
4	Endothrix	negative	*T. soudanense*
5	Endothrix	negative	Negative
6	Endothrix	negative	Negative
7	Ecto-Endothrix	*M. audouinii*	*M. audouinii*
8	Endothrix	negative	*T. soudanense*
9	Endothrix	negative	negative
10	Endothrix	negative	negative
11	Endothrix	negative	*T. soudanense*
12	Endothrix	negative	negative
13	Ecto-Endothrix	*M. canis*	*M. canis*
14	Endothrix	negative	negative
15	Ecto-Endothrix	*M. audouinii*	*M. audouinii*
16	Ecto-Endothrix	*M. audouinii*	*M. audouinii*
17	Endothrix	negative	*T. soudanense*
18	Endothrix	*T. soudanense*	*T. soudanense*
19	Ecto-Endothrix	*M. canis*	*M. canis*
20	Endothrix	negative	negative
21	Endothrix	negative	negative
22	Endothrix	negative	negative
23	Endothrix	*T. soudanense*	*T. soudanense*
24	Endothrix	*T. soudanense*	*T. soudanense*
25	Endothrix	negative	negative
26	Endothrix	*T. soudanense*	*T. soudanense*
27	Endothrix	negative	negative
28	Endothrix	negative	negative
29	Endothrix	negative	negative
30	Ecto-Endothrix	*M. audouinii*	*T. soudanense/M. audouinii*
31	Endothrix	negative	negative
32	Endothrix	negative	negative
33	Endothrix	negative	*T. soudanense*
34	Endothrix	negative	negative
35	Endothrix	negative	*T. soudanense*
36	Endothrix	negative	negative
37	Endothrix	*T. soudanense*	*T. soudanense*
38	Ecto-Endothrix	*M. audouinii*	*M. audouinii*
39	Endothrix	negative	*T. soudanense*
40	Endothrix	negative	negative
41	Endothrix	negative	negative
42	Endothrix	negative	negative
43	Endothrix	negative	negative
44	Endothrix	negative	negative
45	Endothrix	negative	*T. soudanense*
46	Endothrix	negative	*T. soudanense*
47	Endothrix	negative	*T. soudanense*
48	Endothrix	negative	negative
49	Endothrix	negative	negative
50	Endothrix	negative	negative
51	Endothrix	Negative	negative
52	Ecto-Endothrix	*M. audouinii*	*M. audouinii*
53	Endothrix	Negative	negative
54	Endothrix	Negative	*T. soudanense*
55	Ecto-Endothrix	*M. audouinii*	*T. soudanense/M. audouinii*
56	Endothrix	*T. soudanense*	negative
57	Endothrix	Negative	negative
58	Endothrix	Negative	negative
59	Endothrix	Negative	negative
60	Endothrix	Negative	negative
61	Endothrix	Negative	*T. soudanense*
62	Endothrix	*T. soudanense*	*T. soudanense*
63	Endothrix	Negative	negative
64	Endothrix	Negative	negative
65	Endothrix	Negative	negative
66	Endothrix	Negative	negative
67	Endothrix	Negative	negative
68	Endothrix	*T. soudanense*	*T. soudanense*
69	Endothrix	Negative	negative
70	Endothrix	Negative	negative
71	Endothrix	Negative	negative
72	Endothrix	*T. soudanense*	*T. soudanense*
73	Endothrix	Negative	negative
74	Endothrix	Negative	negative
75	Endothrix	Negative	negative
76	Endothrix	*T. soudanense*	*T. soudanense*
77	Endothrix	*T. soudanense*	*T. soudanense*
78	Endothrix	*T. soudanense*	negative
79	Endothrix	T. soudanense	negative
80	Endothrix	Negative	negative
81	Ecto-Endothrix	*M. audouinii*	*M. audouinii*
82	Endothrix	*T. soudanense*	*T. soudanense*
83	Endothrix	*T. soudanense*	negative
84	Endothrix	Negative	negative
85	Endothrix	*T. soudanense*	negative
86	Ecto-Endothrix	*M. audouinii*	*M. audouinii*
87	Endothrix	*T. soudanense*	*T. soudanense*
88	Endothrix	Negative	negative
89	Endothrix	Negative	negative
90	Endothrix	Negative	*T. soudanense*
91	Endothrix	Negative	*T. soudanense*
92	Endothrix	Negative	negative
93	Endothrix	Negative	negative
94	Endothrix	Negative	negative
95	Endothrix	Negative	*T. soudanense*
96	Endothrix	Negative	negative
97	Endothrix	Negative	negative
98	Endothrix	Negative	negative
99	Ecto-Endothrix	*M. audouinii*	*T. soudanense/M. audouinii*
100	Ecto-Endothrix	*M. audouinii*	*M. canis*
101	Endothrix	negative	negative
102	Endothrix	negative	*T. soudanense*
103	Endothrix	negative	*T. soudanense*
104	Endothrix	negative	Negative
105	Endothrix	negative	*T. soudanense*
106	Endothrix	*T. soudanense*	*T. soudanense*
107	Ecto-Endothrix	*M. audouinii*	*M. audouinii*
108	Ecto-Endothrix	*M. audouinii*	*M. audouinii*
109	Endothrix	*T. soudanense*	*T. soudanense*
110	Endothrix	*T. soudanense*	*T. soudanense*
111	Endothrix	*T. soudanense*	*T. soudanense*
112	Ecto-Endothrix	*M. canis*	*M. canis*
113	Endothrix	*T. soudanense*	*T. soudanense*
114	Endothrix	*T. soudanense*	*T. soudanense*
115	Endothrix	*T. soudanense*	*T. soudanense*
116	Ecto-Endothrix	*M. audouinii*	*M. audouinii*
117	Ecto-Endothrix	*M. audouinii*	*M. audouinii*
118	Endothrix	*T. soudanense*	*M. canis*
119	Endothrix	*T. soudanense*	*T. soudanense/M. canis*
120	Endothrix	*T. soudanense*	*T. soudanense/M. audouinii*
121	Ecto-Endothrix	*M. audouinii*	*T. soudanense/M. audouinii*
122	Endothrix	*T. soudanense*	*T. soudanense*
123	Ecto-Endothrix	*M. audouinii*	*T. soudanense*
124	Ecto-Endothrix	*M. audouinii*	*M. audouinii*
125	Endothrix	*T. soudanense*	*T. soudanense*
126	Endothrix	*T. soudanense*	*T. soudanense*
127	Endothrix	*T. soudanense*	*T. soudanense*
128	Endothrix	*T. soudanense*	*T. soudanense*
129	Endothrix	*T. soudanense*	*T. soudanense*

*T. = Trichophyton*, *M. = Microsporum*.

**Table 2 jof-08-00011-t002:** Evaluation of complete multiplex real time PCR DermaGenius^®^ (DG PCR) versus the reference test (fungal culture).

	Culture		Total	Kappa	*p*
PCR DG	Positive	Negative			
Positive	50	18	68		
Negative	6	55	61	0.62	<0.001 **
Total	56	73	129		

Sensitivity (89.3%), Specificity (75.3%), Positive predictive value 73.3%, Negative predictive value 90.2%, Accuracy 81.4%. *p* = Probability. **: highly significance.

**Table 3 jof-08-00011-t003:** Species identified by culture and complete multiplex real-time PCR DermaGenius^®^ (DG PCR) in hair samples (*n* = 129).

Dermatophytes	Culture, *n* (%) [95% CI]	PCR DG, *n* (%) [95% CI]
*T. soudanense*	35 (27.1) [18.63–33.93]	45 (34.9) [26.71–43.77]
*M. audouinii*	18 (14) [8.48–21.15]	12 (9.30) [4.90–15.69]
*M. canis*	3 (2.3) [0.48–6.65]	5 (3.9) (1.27–8.81]
*T. soudanense/M. audouinii*	0	5 (3.9) [1.27–8.81]
*T. soudanense/M. canis*	0	1 (0.78) [0.02–4.24]
Négative	73 (56.6) [47.58–65.29]	61 (47.3) [38.44–56.26]
Total	129 (100) [100]	129 (100) [100]

*T. = Trichophyton*; *M. = Microsporum*; CI = Confidence intervalle.

**Table 4 jof-08-00011-t004:** Summary of studies comparing sensitivity and specificity between real time PCR commercials kits and in-house PCR techniques.

References	Year (Country)	Number and Type of Samples	Sensitivity (%)	Specificity (%)
Commercials kits				
Our study	2020 (Senegal)	129 hairs	89.3	75.3
Hayette et al. (27)	2019 (Belgium)	138 nails	80	74.4
Non commercial kits				
Wisselink et al. (31)	2011 (Netherlands)	1437 (nail, skin and hair)	97	100
Bergman et al. (32)	2013 (Sweden)	202 (152 nail, 44 skins, 5 hair)	99	92
Walser et al. (33)	2019 (Switzerland)	3052 (187 nail, 108 skin, 10 hair)	96.9	90.4

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
