# Peer review of "Evaluation of the Multiplex Real-Time PCR DermaGenius® Assay for the Detection of Dermatophytes in Hair Samples from Senegal"

_jof, 2021, doi:10.3390/jof8010011_

Round 1

Reviewer 1 Report

Thank you for submitting the manuscript for consideration in Journal of Fungi. The study evaluates the performance of DG PCR against the conventional microscopy/culture method for detection of dermatophytes in hair samples. The study would be of interest to dermatologists and pathologists involved in the diagnosis of fungal infections. My comments are as follows:

  1. Abstract – Line 28: Should the sensitivity and specificity be 89.3% and 75.3%, respectively to be consistent with the results in Table 1?
  2. Introduction - The authors may want to provide description of the epidemiology of dermatophytoses in Senegal or the region.
  3. Introduction - The authors have compared how DG PCR performs in relation to other commercially available kits in the market, it would be good to put in a few sentences in the Introduction to explain why the DG PCR was selected for the evaluation over the others.
  4. Methods – The study could have included some negative microscopy and negative culture specimens as “confirmed negative controls”, These will exclude any positive (i.e., false negative) cases that were not detected by both culture and DG PCR. This can be mentioned as a study limitation in discussion.
  5. Methods – Statistical Analysis: How was the sample size determined? Please provide the statistical calculation.
  6. Results – The authors may want to indicate that Figure 1 is the results obtained from DG PCR.
  7. Results – Line 141: Remove ‘The” Figure 1. The sentence should start with “Figure 1…..”.
  8. Results – Table 2: There were 16 soudanense detected by DG PCR, negative by culture. However, Line 167 states 18 instead. Can the authors please clarify?
  9. Results – The authors may want to include a sentence that all there were no specimens tested positive for the other microorganisms in the test panel (i.e., albicans, T. interdigitale, T. mentagrophytes, T. tonsurans, T. violaceum, T. benhamiae, T. verrucosum, and E. floccosum). Please confirm if this is the case.
  10. Discussion – Line 189-194: Can the authors please clarify how was the prevalence established?
  11. Discussion – Line 191: Please remove the extra ‘this’ in the sentence.
  12. Discussion - Line 239: Please cite reference [25] at the end of the sentence.
  13. Discussion – Line 244: Should the sensitivity and specificity be 89.3% and 75.3%, respectively to be consistent with the results in Table 1 and that mentioned elsewhere in the text?
  14. Discussion – Line 249: Please edit the sentence to “The results of the kit have not been compared here with the results obtained from fungal culture,…….”
  15. Discussion – Line 251: Please edit the sentence to “For this study, samples were pre-selected, which included both frequent and less frequent and rare species of dermatophytes contained.
  16. Discussion – Line 252-260: Please edit the sentences to be structurally correct.
  17. Discussion – Line 266: Please edit the sentence to “Other commercial kits which utilize this technology are on the market.”
  18. Discussion – Line 292-298: I have difficulty understanding the context of this paragraph. Please revise the English phrasing.
  19. Discussion – Line 315: The word ‘sloppy’ is not necessary. Please delete.
  20. Discussion – Line 329-345: The authors may want to comment on the usefulness of the DG PCR in speciating and the potential influence this has on therapeutic management.
  21. General: Please use italics when referring to the specie names of the organisms.

Author Response

Response to Reviewer 1 Comments

Dear Reviewers,

We thank all the reviewers of our manuscript whose comments and suggestions will certainly contribute to the improvement of its quality.

All the corrections raised by the different reviewers were directly made in the text (highlight in yellow) and the answers to the different questions and suggestions are mentioned in the table below.

Point 1: Abstract – Line 28: Should the sensitivity and specificity be 89.3% and 75.3%, respectively to be consistent with the results in Table 1?

Response 1: Yes, the sensitivity and specificity will be 89.3% and 75.3%, respectively. Line 28

Point 2: Introduction - The authors may want to provide description of the epidemiology of dermatophytoses in Senegal or the region.

Response 2: We have developed the epidemiology of dermatophytosis (tinea capitis) in lines 145-152

Point 3: Introduction - The authors have compared how DG PCR performs in relation to other commercially available kits in the market, it would be good to put in a few sentences in the Introduction to explain why the DG PCR was selected for the evaluation over the others.

Response 3: The DermaGenius (DG) 2.0 complete multiplex kit (PathoNostics, The Netherlands) is a new commercial real-time polymerase chain reaction (PCR) assay. The evaluation was based on the melting curve analysis. A total of 11 dermatophytes and 1 yeast fungus (Candida albicans) can be identified with the real-time PCR test at the species level. DG 2.0 complete is a very fast and easy to perform test, especially since no post-PCR step is necessary.Line 59-64

Point 4: Methods – The study could have included some negative microscopy and negative culture specimens as “confirmed negative controls”, These will exclude any positive (i.e., false negative) cases that were not detected by both culture and DG PCR. This can be mentioned as a study limitation in discussion.

Response 4: The limitations of our study would be to include some negative microscopy and negative culture specimens as “confirmed negative controls”, These will exclude any positive (i.e., false negative) cases that were not detected by both culture and DG PCR. Line 350-353

Point 5: Methods – Statistical Analysis: How was the sample size determined? Please provide the statistical calculation.

Response 5: there was no initial determination of the sample size. As described in the methods section, all the patients with adequate available sample for DNA extraction were included in this retrospective study.

Point 6: Results – The authors may want to indicate that Figure 1 is the results obtained from DG PCR.

Response 6:  Figure 1 indicate the distribution of Tinea capitis according to age groups

Point 7: Results – Line 141: Remove ‘The” Figure 1. The sentence should start with “Figure 1…..”.

Response 7: ” The” is remove Line 146

Point 8: Results – Table 2: There were 16 soudanense detected by DG PCR, negative by culture. However, Line 167 states 18 instead. Can the authors please clarify?

Response 8: Yes, according Table 2 ,16 T.soudanense were detected by DG PCR . Corrected line 172

Point 9: Results – The authors may want to include a sentence that all there were no specimens tested positive for the other microorganisms in the test panel (i.e., albicansT. interdigitaleT. mentagrophytesT. tonsuransT. violaceumT. benhamiaeT. verrucosum, and E. floccosum).

Response 9: In this study, the others microorganisms in this test panel (i.e., C.albicansT. interdigitaleT. mentagrophytesT. tonsuransT. violaceumT. benhamiaeT. verrucosum, and E. floccosum) were not detected. Lines 175-177

Point 10: Discussion – Lines 189-194: Can the authors please clarify how was the prevalence established?

Response 10: The prevalence was calculated from the culture results

Point 11: Discussion – Line 191: Please remove the extra ‘this’ in the sentence.

Response 11: the extra ‘this’is remove. Line 198

Point 12: Discussion - Line 239: Please cite reference [25] at the end of the sentence.

Response 12: Reference [25] in line 246

Point 13: Discussion – Line 244: Should the sensitivity and specificity be 89.3% and 75.3%, respectively to be consistent with the results in Table 1 and that mentioned elsewhere in the text?

Response 13: Yes, the sensitivity and specificity will be 89.3% and 75.3%, respectively. Line 251

Point 14: Discussion – Line 249: Please edit the sentence to “The results of the kit have not been compared here with the results obtained from fungal culture,…….”

Response 14 : Correted lines 255-256

Point 15: Discussion – Line 251: Please edit the sentence to “For this study, samples were pre-selected, which included both frequent and less frequent and rare species of dermatophytes contained.

Response 15: Correted lines 257-258

Point 16: Discussion – Lines 252-260: Please edit the sentences to be structurally correct.

Response 16: Of 49 samples, the DermaGenius® 2.0 Complete Assay 37 species was detected on specific melting temperature. In 12 samples, one dermatophyte species was detected, but closely related species were not differentiated. Lines 259-261

Point 17: Discussion – Line 266: Please edit the sentence to “Other commercial kits which utilize this technology are on the market.”

Response 17: Corrected line2 67

Point 18: Discussion – Line 292-298: I have difficulty understanding the context of this paragraph. Please revise the English phrasing.

Response 18: I will ask MDPI to correct the English

Point 19: Discussion – Line 315: The word ‘sloppy’ is not necessary. Please delete.

Response 19: Corrected line 317

Point 20: Line 329-345: The authors may want to comment on the usefulness of the DG PCR in speciating and the potential influence this has on therapeutic management.

Response 20: The most plausible explanation for this is that resting fungal cells (e.g. in the form of arthroconidia) are still present and may potentially germinate again after discontinuation of therapy. Dormant cells are missed in the culture. Therefore, therapy control with PCR procedures may be suitable in the future, not to mention the short time-span in which such a finding is available, in order to decide whether to continue the therapy. Only very special PCR procedures are able to discriminate between live and dead cells; however, it is not known how long dormant fungal cells survive in the nail, hair or skin of the human body.

GRÄSER Y. and SAUNTE DML. A Hundred Years of Diagnosing Superficial Fungal Infections: Where Do We Come From, Where Are We Now and Where Would We Like To Go? Acta Dermato-Venereologi

Point 21: General: Please use italics when referring to the specie names of the organisms.

Response 21: Corrected in the paper

Reviewer 2 Report

The manuscript #jof-1498136, entitled “Evaluation of the multiplex real-time PCR DermaGenius® assay for the detection of dermatophytes in hair samples from Senegal” by Ndiaye et al. presents, in line with the title, evaluation on clinical samples of a commercial kit designed for detection and differentation of dermatophytes. In my opinion the study lacks some fundamental data, which I enlist below.

  1. The authors mentioned that the "Positive specimens for dermatophytes were identified following three criteria: growth rate, macroscopic and microscopic characteristics of colonies and sometimes on biochemical characteristics such as urease test." (lines: 86-89). However, in order to show the real time PCR data as convincing, the authors should include a table (biochemical reactions, PCR results etc.) and figure (macromorhology, micromorphology) which will compare all the mentioned charactistics with the data acquired with the kit. I understand that it is a lot of data, however, the authors might present some most notable examples in the main text, and the rest in the Supplementary materials. Additionally, when a detection kit is evaluated on isolates of clinical origin, all the results must be compared with referential strains, such as those from ATCC or DSMZ collections. Additionally, the data must be compared with ITS sequencing. To sum up, the authors should collect all the data, compare it with referential strains and with the detection kit.
  2. The authors mention that "Master Mix 1 (MMX1) contained the originally designed specific PCR primer pairs and detection probes for C. albicans, T. interdigitale, T. mentagrophytes, T. tonsurans, T. violaceum, and T. rubrum/soudanense, and MMX2 contained the originally designed primer pairs and probes for T. benhamiae, T. verrucosum, M. canis, M. audouinii, and E. floccosum." (lines 108-112). This type of presentation is not convincing. The authors must include the exact information on the primers and probes. If such data is a company secret of the manufacturer, thus all the results acquired with the kit cannot be considered convincing and (in my opinion) are not meant to be published. Additionally, it is worth comparing the Pairwise Sequence Alignment of the designed DNA sequences between species included in one of the MIX. Such comaprison will allow understanding how and why the kit differentiates between different fungal species in one reaction. 

Author Response

Response to Reviewer 2 Comments

Dear Reviewers,

We thank all the reviewers of our manuscript whose comments and suggestions will certainly contribute to the improvement of its quality.

All the corrections raised by the different reviewers were directly made in the text (highlight in yellow) and the answers to the different questions and suggestions are mentioned in the table below.

Point 1: The authors mentioned that the "Positive specimens for dermatophytes were identified following three criteria: growth rate, macroscopic and microscopic characteristics of colonies and sometimes on biochemical characteristics such as urease test." (lines: 86-89). However, in order to show the real time PCR data as convincing, the authors should include a table (biochemical reactions, PCR results etc.) and figure (macromorhology, micromorphology) which will compare all the mentioned charactistics with the data acquired with the kit. I understand that it is a lot of data, however, the authors might present some most notable examples in the main text, and the rest in the Supplementary materials. Additionally, when a detection kit is evaluated on isolates of clinical origin, all the results must be compared with referential strains, such as those from ATCC or DSMZ collections. Additionally, the data must be compared with ITS sequencing. To sum up, the authors should collect all the data, compare it with referential strains and with the detection kit.

Response 1: These characteristics are used to identify dermatophytes. they are not listed in the registers. they are displayed and used to discriminate between two species

Point 2 The authors mention that "Master Mix 1 (MMX1) contained the originally designed specific PCR primer pairs and detection probes for C. albicans, T. interdigitale, T. mentagrophytes, T. tonsurans, T. violaceum, and T. rubrum/soudanense, and MMX2 contained the originally designed primer pairs and probes for T. benhamiae, T. verrucosum, M. canis, M. audouinii, and E. floccosum." (lines 108-112). This type of presentation is not convincing. The authors must include the exact information on the primers and probes. If such data is a company secret of the manufacturer, thus all the results acquired with the kit cannot be considered convincing and (in my opinion) are not meant to be published. Additionally, it is worth comparing the Pairwise Sequence Alignment of the designed DNA sequences between species included in one of the MIX. Such comaprison will allow understanding how and why the kit differentiates between different fungal species in one reaction. 

Response 2: Yes data is a company secret of the manufacturer. DG is a Certified European in vitro diagnostic assay (CE-IVD), which targets the internal transcribed spacer (ITS) region of the ribosomal fungal genome.

Reviewer 3 Report

The authors performed a study on 159 patients with th eclinical suspect diagnosis of tinea capitis. They found that a commercial real-time PCR was more sensitive and specific than culture. This is in accordance with virtually all previous studies.

The discussion is very wordy and would benefit from considerable shortening.

"... preceded by the use of Wood’s 70 lamp which orient to certain species such as M. canis and M. audouinii. "  Correct to "... preceded by the use of Wood’s 70 lamp which orients to certain species such as M. canis and M. audouinii."

"...  DNA isolation from hairs samples."  Correct to "... DNA isolation from hair samples. "

"The patients between age group 0-10 years had ..." You probably mean: "The patients in the age group between 0 and 10 years had .."

"... than culture for dermatophytes detection ..." Correct to "... than culture for dermatophyte detection ..."

"... children for 6 months to 10– 12 years of age."  Correct to "...children of 6 months to 10– 12 years of age. "

"...the culture of nail samples yielded to Microsporum langeronii..." Omit "to"

"Other commercial kits, who are interested in this technology are on the market." Correct to "Other commercial kits, which are interested in this technology are on the market."

"The Kit contains only one universal dermatophyte detection, the differentiation of However, genera or species shall not be possible." Correct to "The Kit contains only one universal dermatophyte detection, however, the differentiation of genera or species shall not be possible."

"A study from the year 2016 proves the superiority of the method in terms of the sensitivity in comparison to the culture, in particular in the case of authors so-called non-optimal samples, where the distal, outgrowing images nails have been removed and not on Transition from diseased to healthy nails (optimal) 273 was sampled[29]." Please rephrase as this is difficult to understand.

"In 2019, Uhrlab et al were compared DG PCR and EuroArray tests regarding their 280 diagnostic specificity to identify dermatophyte DNA." Omit "were".

"... less dermatophyte species as the EuroArray ..." Correct to "... less dermatophyte species than the EuroArray ..."

"My study shows that .." Not "Our study shows that ..."?

".... one of the mainly dermatophyte implicated in tinea capitis ..." Correct to "... one of the main dermatophytes implicated in tinea capitis ..."

"The requirement of this test, which is used in the daily. For DG PCR, the distinction ..." Do you mean "The requirement of this DG PCR test, which is used in daily practice (or routine) for the distinction ..."?

"T. soudanense and T. rubrum will be informed about common proof is covered." I do not understand what exactly you mean; please rephrase.

"These dermatophytes can be genotypically similar from each other." They  can be genotypically similar to each other!

"DG PCR showed excellent performance characteristics ... is significantly faster than cultures techniques, which makes it very promising for routine diagnostics of dermatophytosis in Africa particularly in Senegal." I do not understand: The DG PCR was performed in Belgium!? 

The references are mostly not complete: volume numbers and pages are lacking, often authors are not complete, spelling is inconsistent.

Author Response

Response to Reviewer 3 Comments

Dear Reviewers,

We thank all the reviewers of our manuscript whose comments and suggestions will certainly contribute to the improvement of its quality.

All the corrections raised by the different reviewers were directly made in the text (highlight in yellow) and the answers to the different questions and suggestions are mentioned in the table below.

Point 1: "... preceded by the use of Wood’s 70 lamp which orient to certain species such as M. canis and M. audouinii. " Correct to "... preceded by the use of Wood’s 70 lamp which orients to certain species such as M. canis and M. audouinii."

Response 1: Corrected lines 75-76

Point 2: "...  DNA isolation from hairs samples."  Correct to "... DNA isolation from hair samples.

Response 2: Corrected line 97

Point 3: "The patients between age group 0-10 years had ..." You probably mean: "The patients in the age group between 0 and 10 years had .."

Response 3: Corrected line 150

Point 4: "... than culture for dermatophytes detection ..." Correct to "... than culture for dermatophyte detection ..."

Response 4: Corrected lines 161

Point 5: "... children for 6 months to 10– 12 years of age."  Correct to "...children of 6 months to 10– 12 years of age. "

Response 5: Corrected line 215

Point 6: "...the culture of nail samples yielded to Microsporum langeronii..." Omit "to"

Response 6: Corrected lines 237

Point 7: "Other commercial kits, who are interested in this technology are on the market." Correct to "Other commercial kits, which are interested in this technology are on the market."

Response 7: Corrected line 267 by Reviewer 1

Point 8: "The Kit contains only one universal dermatophyte detection, the differentiation of However, genera or species shall not be possible." Correct to "The Kit contains only one universal dermatophyte detection, however, the differentiation of genera or species shall not be possible."

Response 8: Corrected lines 271-272

Point 9: "A study from the year 2016 proves the superiority of the method in terms of the sensitivity in comparison to the culture, in particular in the case of authors so-called non-optimal samples, where the distal, outgrowing images nails have been removed and not on Transition from diseased to healthy nails (optimal) 273 was sampled[29]." Please rephrase as this is difficult to understand.

Response 9: A study from the year 2016 proves the superiority of the molecular method in terms of the sensitivity in comparison to the culture.This study is the first retrospective evaluation of BioEvolution’s real-time PCR kit, carried out on 180 nails, divided into optimal and non-optimal samples.When comparing the number of dermatophytes found by culture and the molecular method, a larger number of dermatophytes was detected with this molecular kit:only 23.3% (21/90) and 16.7% (15/90) respectively of the optimal and non-optimal samples, obtained from the same patients, were found positive in culture, whereas the PCR resulted in 34.4% (31/90) of positive cases whatever the sample quality.Lines 272-280

Point 10: "In 2019, Uhrlab et al were compared DG PCR and EuroArray tests regarding their 280 diagnostic specificity to identify dermatophyte DNA." Omit "were".

Response 10: Corrected line 286

Point 11: "... less dermatophyte species as the EuroArray ..." Correct to "... less dermatophyte species than the EuroArray ..."

Response 11: Corrected lines 292-293

Point 12: "My study shows that .." Not "Our study shows that ..."?

Response 12: Corrected line 295

Point 13: ".... one of the mainly dermatophyte implicated in tinea capitis ..." Correct to "... one of the main dermatophytes implicated in tinea capitis ..."

Response 13: Corrected line 296

Point 14: "The requirement of this test, which is used in the daily. For DG PCR, the distinction ..." Do you mean "The requirement of this DG PCR test, which is used in daily practice (or routine) for the distinction ..."?

Response 14: We mean to say “The requirement of this DG PCR test, which is used in daily practice (or routine) for the distinction…..lines 298-299

Point 15: "T. soudanense and T. rubrum will be informed about common proof is covered." I do not understand what exactly you mean; please rephrase.

Response 15: T. soudanense and T. rubrum will not be differenttly identified.

Point 16: "These dermatophytes can be genotypically similar from each other." They  can be genotypically similar to each other!

Response 16: They can be genotypically similar to each other. Line 336

Point 17: "DG PCR showed excellent performance characteristics ... is significantly faster than cultures techniques, which makes it very promising for routine diagnostics of dermatophytosis in Africa particularly in Senegal." I do not understand: The DG PCR was performed in Belgium!? 

Response 17: Indeed, the DG PCR was performed in Belgium. I would like to say that due to its performance DGPCR could be an alternative for the diagnosis of dermatophytes in Senegal.

Round 2

Reviewer 1 Report

Thank you for taking careful consideration of the comments in your revision of the manuscript. The revised version has been significantly improved.

Please edit Line 175-176 by changing 'others microorganisms' to 'other microorganisms'.

Please edit the typographical error in Line 302 'differently'.

The language of the article should be further improved. Please seek professional English editing service or someone with English as native language. 

Author Response

Dear Reviewers,

We thank all the reviewers of our manuscript whose comments and suggestions will certainly contribute to the improvement of its quality.

All the corrections raised by the different reviewers were directly made in the text (highlight in yellow) and the answers to the different questions and suggestions are mentioned in the table below.

Point 1: Please edit Line 175-176 by changing 'others microorganisms' to 'other microorganisms'.

Response 1: Corrected line 175-176

Point 2: Please edit the typographical error in Line 302 'differently'.

Response 2: Corrected line 175-176

Point 3: The language of the article should be further improved. Please seek professional English editing service or someone with English as native language. 

Response 3: The manuscript will be send to MDPI for extensive English revisions.

Reviewer 2 Report

The manuscript #jof-1498136, entitled “Evaluation of the multiplex real-time PCR DermaGenius® assay for the detection of dermatophytes in hair samples from Senegal” by Ndiaye et al. has been revised. In my opinion the study still lacks some fundamental data, which I enlist during first round of the review.

The manuscript, which describes the studies in the title as "evaluation" should evaluate the usefulness of the commercial kit in accordance with the scientific art. The proper detection of the isolates by the kit must be double-check with already known methods (classical mycological evaluation + molecular techniques). Also, isolates of clinical origin should be compared to referential strains. Additionally, to present the data as convincing (and reproductive) all the necessary information on the mechanism of detection by the kit must be included in the manuscript.

I believe that including both points in the manuscript would meet both criteria - the reproducibility and the transparency.

Author Response

Dear Reviewers,

We thank all the reviewers of our manuscript whose comments and suggestions will certainly contribute to the improvement of its quality.

All the corrections raised by the different reviewers were directly made in the text (highlight in yellow) and the answers to the different questions and suggestions are mentioned in the table below.

Point 1: The manuscript, which describes the studies in the title as "evaluation" should evaluate the usefulness of the commercial kit in accordance with the scientific art. The proper detection of the isolates by the kit must be double-check with already known methods (classical mycological evaluation + molecular techniques). Also, isolates of clinical origin should be compared to referential strains. Additionally, to present the data as convincing (and reproductive) all the necessary information on the mechanism of detection by the kit must be included in the manuscript.

I believe that including both points in the manuscript would meet both criteria - the reproducibility and the transparency.

Response 1: According to the manufacturer’s recommendations, internal control (IC) was added to each sample to monitor DNA-extraction performance. A positive control and negative template control (NTC) were included in each PCR run. This has been explained in the section 2.3 and 2.4. For more information, please see the article by Hayette & al who used the same KIT.

  • Hayette, M.P.; Seidel, L.; Adjetey, C.; Darfouf, R.; Wéry, M.; Boreux, R.; Sacheli, R.; Melin, P.; Arrese, J. Clinical evaluation of the DermaGeniusR Nail real-time PCR assay for the detection of dermatophytes and Candida albicans in nails. Mycol. 2019, doi:10.1093/mmy/myy020.

Reviewer 3 Report

The manuscript has been corrected and linguistically improved.

Author Response

please see the attachement
